# Exogenous Iron Increases Fasciocidal Activity and Hepatocellular Toxicity of the Synthetic Endoperoxides OZ78 and MT04

**DOI:** 10.3390/ijms20194880

**Published:** 2019-10-01

**Authors:** Karin Brecht, Carla Kirchhofer, Jamal Bouitbir, Francesca Trapani, Jennifer Keiser, Stephan Krähenbühl

**Affiliations:** 1Division of Biopharmacy, Department of Pharmaceutical Sciences, University of Basel, CH-4056 Basel, Switzerland; Karin.Brecht@unibas.ch; 2Department of Medical Parasitology and Infection Biology, Swiss Tropical and Public Health Institute, University of Basel, CH-4002 Basel, Switzerland; carla.kirchhofer@hotmail.com (C.K.);; 3Division of Clinical Pharmacology & Toxicology, Department of Medicine, University of Basel, CH-4031 Basel, Switzerland; jamal.bouitbir@unibas.ch; 4Department of Biomedicine, University of Basel, CH-4031 Basel, Switzerland; 5Swiss Centre of Applied Human Toxicology (SCAHT), University of Basel, CH-4001 Basel, Switzerland; 6Institute of Pathology, University of Basel, CH-4003 Basel, Switzerland; francescatrapani.vet@gmail.com

**Keywords:** Fasciola hepatica, artesunate, OZ78, MT04, HepG2 cells, hepatotoxicity

## Abstract

The synthetic peroxides OZ78 and MT04 recently emerged as fasciocidal drug candidates. However, the effect of iron on fasciocidal activity and hepatocellular toxicity of these compounds is unknown. We investigated the in vitro fasciocidal activity and hepatocellular toxicity of OZ78 and MT04 in absence and presence of Fe(II)chloride and hemin, and conducted a toxicological study in mice. Studies were performed in comparison with the antimalarial artesunate (AS), a semisynthetic peroxide. Fasciocidal effects of OZ78 and MT04 were confirmed and enhanced by Fe^2+^ or hemin. In HepG2 cells, AS reduced cellular ATP and impaired membrane integrity concentration-dependently. In comparison, OZ78 or MT04 were not toxic at 100 µM and reduced the cellular ATP by 13% and 19%, respectively, but were not membrane-toxic at 500 µM. The addition of Fe^2+^ or hemin increased the toxicity of OZ78 and MT04 significantly. AS inhibited complex I, II, and IV of the mitochondrial electron transport chain, and MT04 impaired complex I and II, whereas OZ78 was not toxic. All three compounds increased cellular reactive oxygen species (ROS) concentration-dependently, with a further increase by Fe^2+^ or hemin. Mice treated orally with up to 800 mg OZ78, or MT04 showed no relevant hepatotoxicity. In conclusion, we confirmed fasciocidal activity of OZ78 and MT04, which was increased by Fe^2+^ or hemin. OZ78 and MT04 were toxic to HepG2 cells, which was explained by mitochondrial damage associated with ROS generation in the presence of iron. No relevant hepatotoxicity was observed in mice in vivo, possibly due to limited exposure and/or high antioxidative hepatic capacity.

## 1. Introduction

*Fasciola hepatica* (*F. hepatica*) is a food-borne trematode, which infects 2 to 17 million people worldwide. The parasite settles in the liver, bile ducts, and the gallbladder of the host, causing fascioliasis. A global burden of about 1.7 million DALYs (disability-adjusted life years) has been attributed to the cluster of food-borne trematode infections in 2016 [1,2,3]. Fascioliasis is not only an important human health problem but is also of enormous importance in veterinary health [4].

Triclabendazole is the first-choice therapy against fascioliasis. However, resistance to this anthelmintic drug has been documented globally in veterinary medicine and is threatening treatment and control of fascioliasis [5].

Artemisinin (see Figure 1 for chemical structure), a natural product from the plant *Artemisia annua*, and its derivatives are the most widely used drugs against malaria [6]. The semi-synthetic, water-soluble derivative artesunate (AS) and its active metabolite α-β-dihydroartemisinin (DHA) are also active against the liver fluke *F. hepatica* [7]. In comparison to the semisynthetic artemisinins, fully synthetic compounds, such as trioxolanes and tetraoxanes [8,9], have a higher bioavailability and a higher systemic exposure compared to artesunate [10]. For example, OZ78 and MT04 (Figure 1) exhibited a high trematocidal activity in in vitro and in in vivo models, in parallel to favorable pharmacokinetic properties [11]. In addition, MT04 showed an egg count reduction by 98.5% and a worm burden reduction by 92% in sheep naturally infected with *F. hepatica*, while OZ78 showed no effect in sheep [12,13].

Different studies have shown that the artemisinins and fully synthetic endoperoxides, such as trioxolanes and tetraoxanes, react with iron. Free Fe^2+^ [14,15] or Fe^2+^ in heme [14,16,17,18] reduces and opens the endoperoxide group of these molecules, eventually leading to the formation of C-centered radicals that can react with different cell constituents, such as heme, phospholipids, proteins, and DNA [14,17,19]. It has been proposed that heme produced by the degradation of hemoglobin by malaria parasites is responsible for the activation and therefore, also for the pharmacological activity of artemisinins and synthetic endoperoxides against malaria [20,21]. Heme–artemisinin adducts have been isolated from urine and the spleen of parasite-infected mice but not in uninfected mice treated with artemisinin [22]. Alkylation of heme and of other cell constituents may also be important for the fasciocidal activity of artemisinins and of synthetic endoperoxides.

The endoperoxide group of the artemisinin-based compounds not only acts as a pharmacophore, but it has also been shown to be responsible for neuro- and embryotoxicity in rats and rabbits and to induce mitochondrial dysfunction and apoptosis in HeLa cells [17,23]. Toxic effects were also observed in *F. hepatica* infected rats treated with an oral dose of 400 mg/kg AS [7]. However, due to the poor pharmacokinetic properties of the artemisinins, only low systemic plasma concentrations are reached, and this might be the reason why there are no reports of neuro- or embryotoxicity in man [24]. OZ78 has no mutagenic potential [8], but, to the best of our knowledge, there is no further data published about the toxicity of OZ78 and MT04.

In the current study, we had two principal aims. First, we wanted to elucidate the effect of free Fe^2+^ and heme-complexed iron on the in vitro fasciocidal activity of the synthetic endoperoxides OZ78 and MT04. Second, taking into account the hepatic localization of liver flukes, we attempted to study the toxicity of these compounds on HepG2 cells (a human hepatocarcinoma cell line) and on liver morphology and function of mice in vivo.

## 2. Results

### 2.1. Fe^2+^ and Hemin Potentiate Fasciocidal Activity of OZ78 and MT04 in Worms Ex Vivo

We have previously shown that OZ78 and MT04 (at 50 µg/mL) have activity against the liver fluke *F. hepatica* [25]. In our current work, we investigated temporal effects of OZ78 and MT04 in combination with Fe^2+^ or hemin on adult *F. hepatica* from bovine origin in vitro.

Control worms incubated with 1% DMSO for 72 h did not show a change in viability. In the presence of iron or hemin for 48 h or 72 h, however, *F. hepatica* showed a slight reduction in viability (Figure 2A).

AS (50 µg/mL) was the most active drug tested in this assay against *F. hepatica*. All treatments led to the death of the worms within 72 h (Figure 2B). The combination with iron or hemin did not increase the activity of AS against *F. hepatica* as compared to treatment with AS alone.

Incubation with OZ78 (50 µg/mL) for 24 h had no effect on worm vitality, whereas incubations of OZ78 in combination with iron or hemin resulted in a reduction in viability to score 3 (Figure 2C). Similarly, after 48 h, combined treatment of OZ78 with iron or hemin was more efficient in impairing worm viability than OZ78 alone. After 72 h, however, we observed that all three treatments reached a comparable reduction in movements of *F. hepatica* (score 2).

Worms incubated with MT04 alone were not affected after 24 h. However, after 24 h of coincubation of MT04 (50 µg/mL) with iron or hemin, liver flukes showed a marked reduction in movement (viability score of 2.2 or 2.5, respectively, Figure 2D). After 72 h of treatment with MT04 alone or in combination with iron, we observed reduced viability of *F. hepatica* (score 1.6), whereas MT04 plus hemin led to the death of the worms.

Our results confirm that both OZ78 and MT04 are active drugs against *F. hepatica* and that, in contrast to AS, Fe^2+^ or hemin accelerated or enhanced the toxicity of OZ78 or MT04 (with hemin being more effective than iron).

### 2.2. Impact of AS, OZ78 or MT04 on Membrane Integrity and ATP Content of HepG2 Cells

AS has been described in case reports to be hepatotoxic in humans [26,27,28]. We wanted to corroborate the effect of AS in a human hepatocarcinoma cell line and to extend our studies on the new drug candidates OZ78 and MT04. Having shown that iron and hemin augmented the efficacy of OZ78 and MT04 in worms ex vivo (Figure 2), we included FeCl_2_ and hemin in our toxicological studies (Figure 3).

To investigate membrane integrity, we monitored the release of adenylate kinase induced by the test compounds alone or in combination with Fe^2+^ or hemin (Figure 3A). Triton X significantly increased the adenylate kinase (AK) release by a factor of seven compared to DMSO control (data not shown). Due to its brown color, hemin interfered with the adenylate kinase assay (and all other assays currently available to measure membrane leakage) and could not be evaluated.

AS was increasingly toxic at 100 and 500 µM, with an increase in AK release by 190% and 270%, respectively, as compared to DMSO control. At 100 µM, Fe^2+^ augmented membrane leakage of AS (320%), whereas Fe^2+^ alone was not cytotoxic. High concentrations of AS (500 µM) in the presence of Fe^2+^ accelerated cellular impairment as observed by microscopic investigation (lyzed cells, pictures not shown). However, possibly due to the limited half-life of AK in the supernatant, the significant decrease in viability did not translate in appropriate levels of active AK.

OZ78 or MT04 did not impair membrane integrity up to 500 µM. While the combined treatment of MT04 with Fe^2+^ induced a significant release of AK (200% increase as compared to DMSO), cotreatment of OZ78 with Fe^2+^ did not.

We expanded the cytotoxicity experiments by measuring cellular ATP content (Figure 3B), a non-destructive assay that was not hampered by the hemin solution. Fe^2+^ and hemin alone had no impact on cellular ATP content. In the presence of Triton X as a positive control, almost no ATP was left in cells (data not shown). AS (100 µM) decreased the cellular ATP content by 55% as compared to DMSO after 24 h of incubation. When AS was coincubated with iron or hemin, ATP dropped by 93% and 95%, respectively, as compared to control. Five hundred micromolar of AS alone caused a complete depletion in cellular ATP irrespective of the presence of iron or hemin.

OZ78 (100 μM) alone or in combination with Fe^2+^ was not toxic. In contrast, cotreatment of OZ78 with hemin triggered a slight, but significant drop in ATP (12%). At 500 μM, OZ78 alone showed a minor toxic effect (14% decrease in ATP), which was not enhanced by the addition of iron or hemin.

MT04 at 100 μM was not toxic for HepG2 cells. However, in combination with iron or hemin, a drop in the ATP content by 20% or 40%, respectively, was detected. High concentrations of MT04 (500 μM) decreased intracellular ATP by 20%. Coincubations with iron or hemin triggered an additional loss in ATP, resulting in an overall decrease of 65%.

The results show that AS was more toxic than the two synthetic peroxides under investigation, with MT04 being more toxic than OZ78 in HepG2 cells. The presence of iron or hemin augmented the cytotoxic effect induced by AS, OZ78, and MT04.

### 2.3. AS and MT04 Inhibit Mitochondrial Respiration

In hepatocytes, the main production of ATP occurs in the mitochondria through oxidative phosphorylation. Having observed a significant drop in cellular ATP levels, we intended to investigate the effect AS 100 µM, MT04 500 µM, and OZ78 500 µM on mitochondrial respiration in HepG2 cells (Figure 4).

Using glutamate and malate as substrates, which produce nicotinamide adenine dinucleotide (NADH) that enters the electron transport chain at complex I, significant inhibition of the maximal ADP-stimulated respiration rate was observed for AS and MT04 treated cells (inhibition of 75% and 53%, respectively) as compared to DMSO.

Using succinate (and rotenone to inhibit complex I) as substrate, which produces flavin adenine dinucleotide (FADH) that enters the electron transport chain at complex II, we again detected significant differences among AS and MT04 treated cells compared to the control. AS inhibited mitochondrial respiration by 83%, whereas MT04 inhibited succinate metabolism by 61% compared to control.

In the presence of N,N,N′,N′-tetramethyl-p-phenylendiamine (TMPD)/ascorbate (and antimycin A to inhibit complex III), which was used to investigate the activity of complex IV, the maximal ADP-stimulated respiration rate was decreased by AS by 47%, whereas MT04 inhibited mitochondrial respiration of HepG2 cells by 22% without reaching statistical significance.

For comparison, OZ78 inhibited mitochondrial oxidative metabolism in the presence of succinate or ascorbate/TMPD only numerically, but without reaching statistical significance.

### 2.4. Effect of AS, OZ78 or MT04 on ROS Accumulation

Impairment of mitochondrial respiration has been associated with the generation of reactive oxygen species (ROS) [29]. Moreover, the generation of radicals in the presence of iron or hemin is a widely accepted explanation of why artemisinins are toxic to parasites and mammalian cells [10,17,19,30]. Therefore, we determined the ROS concentration in HepG2 cells after 90 min and 24 h treatment with AS, OZ78, or MT04 alone or in the presence of iron or hemin (Figure 5).

The solvent control, DMSO, and Fe^2+^ did not trigger ROS production within 90 min or 24 h incubation, whereas hemin induced a significant increase in ROS by a factor of 10 after 90 min or of 30 after 24 h compared to DMSO. In the absence of exogenous iron, none of the compounds investigated increased the cellular ROS after 90 min, but AS and OZ78 increased the cellular ROS content after 24 h by a factor of 5. After an incubation of 90 min, the addition of Fe^2+^ to the compounds investigated did not increase the ROS content. In contrast, the addition of hemin significantly increased the ROS content after 90 min for all compounds investigated compared to respective incubations containing only hemin. After 24 h, the addition of Fe^2+^ increased the cellular ROS content for AS and OZ78, but not for MT04. In comparison, the addition of hemin increased the ROS content for all compounds investigated compared to the respective incubations containing only hemin.

### 2.5. No Significant Metabolism of OZ78 and MT04 by Human Liver Microsomes

It has been reported that AS is extensively metabolized in the liver, inhibits several cytochrome P450 isoforms (CYPs) and auto-induces its own metabolism [31]. For OZ78 and MT04, no data exist regarding their stability in the presence of CYPs. We, therefore, tested the decay of AS, OZ78 or MT04 by measuring the concentration of the respective parent substance after 2 hours of incubation in the presence of functional or heat-inactivated microsomes. We confirmed the rapid and almost complete metabolism of AS after 2 hours of incubation with functional microsomes (94% of AS was metabolized). In contrast, in the presence of heat-inactivated microsomes, no degradation was detected. OZ78 and MT04 did not show a decrease in the presence of functional microsomes, indicating that OZ78 and MT04 are not or very slowly metabolized by CYPs.

### 2.6. No Significant Liver Toxicity of OZ78 and MT04 in Mice

In previous studies, we have shown that oral application of a single dose of AS (200 mg/kg), OZ78 (100 mg/kg) or MT04 (50 mg/kg) to rats infected with *F. hepatica* efficiently killed juvenile and adult liver flukes in vivo [25,32]. In the present study, we focused on potential hepatic adverse effects of AS, OZ78, and MT04 in mice. As AS was found to be the most toxic compound tested in our in vitro toxicity tests in HepG2 cells, we performed a pilot-study treating mice with a single oral dose of 400 mg/kg AS. We analyzed plasma aspartate aminotransferase (AST), alkaline phosphatase (aPhos), and total serum bilirubin after 2, 4, 8, 12, 24 h. Mice were sacrificed after 24 h, and liver biopsies were investigated. Highest values for all clinical parameters tested were obtained at 8 h, and no liver lesions were observed after 24 h (data not shown).

Based on these results, we investigated the effects of increasing doses of AS, OZ78 or MT04 (200–800 mg/kg) in mice 8 hours post-treatment (Figure 6). In the group of AS treated mice, we found that doses of 200 and 400 mg/kg were not associated with liver injury. At 800 mg/kg, one mouse was without lesions, one with steatosis, and one with confluent necrosis (Appendix A). While aPhos and bilirubin did not increase in the group of AS treated mice, AST moderately increased with escalating doses of AS, compatible with hepatocyte necrosis found in liver histology (Figure 6).

In all three groups of mice treated with OZ78, we observed two mice (out of four) with slight hydropic degeneration of hepatocytes (Appendix A). Hydropic degeneration or swelling usually results from non-specific cell injury, leading to water accumulation and is reversible. Alternatively, it may also occur during sample preparation. The three clinical parameters investigated (AST, total bilirubin, and aPhos) were in the normal range (Figure 6).

Mice treated with 200 or 400 mg/kg MT04 also showed no pathologic liver histology (Appendix A). Three out of four mice treated with 800 mg/kg MT04 presented hydropic degeneration. One of these mice had a concomitant elevation of AST. Importantly, we observed a similarly elevated AST activity for a mouse without hydropic degeneration, suggesting that hydropic degeneration was not necessarily associated with hepatocellular damage and may have been an artifact occurring during sample preparation. One mouse with hydropic degeneration had an increased total bilirubin plasma concentration, whereas the other two mice with hydropic degeneration had normal plasma bilirubin values. Two mice treated with 200 mg/kg MT04 had highly elevated AST values, however, without underlying liver pathology. With increasing doses of MT04, we did not observe a concomitant rise in AST (Figure 6), indicating that, in contrast to AS, MT04 did not affect AST levels in a concentration-dependent fashion.

## 3. Discussion

Fascioliasis is a major public health and veterinary problem [33]. Resistance to the only available drug triclabendazole is a big motivation to search for new drugs [34,35]. OZ78 and MT04 have been shown to be effective against fascioliasis [25]. However, there is currently no data available regarding toxicity, and there are knowledge gaps in the mechanism of action.

Our results confirm that both OZ78 and MT04 are active against *F. hepatica* and that the addition of iron, in particular in the form of hemin, enhances both trematocidal activity and toxicity of OZ78 and MT04 (Figure 2). This finding is in line with previous in vitro studies demonstrating augmented tegumental changes caused by OZ78 in the presence of hemin [36]. In addition, in vivo studies demonstrated that nonperoxidic OZ derivatives lack fasciocidal activity in the rat model, strongly supporting a peroxide-bond-dependent mechanism of action against liver flukes [37].

It has been reported that AS is extensively metabolized in the liver, inhibits several cytochrome P450 isoforms (CYPs) and auto-induces its own metabolism [27]. For OZ78 and MT04, no data exist regarding their stability in the presence of CYPs. We, therefore, tested the decay of AS, OZ78 or MT04 in the presence of functional human liver microsomes (hLM) and confirmed the rapid and almost complete degradation of AS. This observation is in line with studies performed in the presence of rat or human microsomes as well as recombinant CYP450s (rCYPs). Previous studies demonstrated that AS is metabolized by CYP2A6, predicting hepatic metabolism and also the potential for drug–drug interactions in vivo [31,38]. In contrast to AS, we did not observe microsomal degradation of OZ78 and MT04 in vitro, suggesting that hepatic metabolism is not necessary for the trematocidal activity of OZ78 and MT04 and that OZ78 and MT04 carry a low risk for drug–drug interactions.

AS has been described in case reports to be hepatotoxic in humans [26,27,28]. The fully synthetic compounds under investigation, OZ78 and MT04, are chemically related peroxides and could also be hepatotoxic, even if they are not metabolized by CYPs. We aimed to investigate whether OZ78 and MT04 reveal hepatocellular toxicity at the concentrations required for fasciocidal activity (100–500 µM). The investigations were also carried out in the presence of two iron compounds, Fe^2+^ and hemin. Peroxide bond cleavage induced by iron, resulting in radical formation, is not only a proposed mechanism of action but could also be associated with toxicity [10,17,30]. The antimalarial drug AS served as a comparator.

We applied two assays to assess cell viability and metabolic toxicity. For that, we chose the AK release assay to measure membrane leakage and the CellTiter Glo assay to assess the cellular ATP content. OZ78 and MT04, at both concentrations tested, did not result in a significant loss in membrane integrity in HepG2 cells. In the presence of Fe^2+^, OZ78 did not result in membrane leakage, but MT04 was slightly toxic. For comparison, the effect on the cellular ATP content was more pronounced for both compounds. OZ78 decreased the cellular ATP content at 500 µM irrespective of the presence of iron or hemin, and MT04 reduced the cellular ATP content to an even greater extent. The higher toxicity of MT04 compared to OZ78 might be explained by the fact that MT04 contains two peroxide bridges as compared to one for OZ78 (Figure 1). AS was the most toxic drug tested. AS showed significant cellular toxicity, in particular in combination with iron, as indicated by AK release and decrease in the cellular ATP content. The in vitro toxicity findings are in line with recent studies, which have shown the occurrence of apoptotic processes in the presence of ferrous salts of the artemisinins on rapidly proliferating cells [30,39].

The observation that the drugs investigated had a more pronounced effect on the cellular ATP content than on membrane integrity (Figure 3) prompted us to assume mitochondrial toxicity and to study the action of the compounds on mitochondria in more detail. As shown in Figure 4, AS inhibited TMPD/ascorbate-mediated oxygen consumption by HepG2 cells, indicating that AS is a complex IV inhibitor. In addition, it impaired glutamate/malate- and succinate-mediated oxygen consumption, compatible with inhibition of complex I and II. Similarly, MT04 inhibited glutamate/malate- and succinate-mediated oxygen consumption, but without inhibiting complex IV. In contrast, OZ78 did not significantly impair mitochondrial function. These data correlate well with ATP depletion in the absence of Fe^2+^ or hemin (Figure 3B). Mitochondrial toxicity may be a consequence of increased cellular ROS levels, which may first increase due to the formation of radicals from the endoperoxides investigated and then also due to the dysfunction of the mitochondrial electron transport chain [17,29]. In support of this assumption, all three compounds investigated were associated with an increase in the cellular ROS content and the addition of iron not only increased ROS formation but also cytotoxicity.

Compatible with impairment of respiratory chain complexes, ATP depletion and AK release induced by AS in cellular assays, we demonstrated that a single high dose of AS gave rise to slightly increased AST levels and concomitant hepatocyte necrosis/steatosis in vivo. Published LD_50_ values of AS in mice were 520 and 475 mg/kg after single intravenous and intramuscular administration, respectively [12]. In our study, we administered up to 800 mg/kg orally with moderate toxicity, indicating that the oral bioavailability in mice was limited. In contrast to AS, we detected no relevant hepatic toxicity for OZ78 or MT04 in vivo, despite the demonstration of clear in vitro toxicity at least for MT04. A possible explanation for this apparent discrepancy is limited exposure. Since we did not measure plasma or liver concentrations, we do not know whether the toxic concentrations were reached in the mice treated with these compounds. It is also possible that the generation of mitochondrial damage needs more time than the eight hours chosen in the current study. Furthermore, the liver has a high antioxidative capacity, which may have suppressed ROS accumulation.

Since resistance is a problem in the treatment of fascioliasis [5], drug combinations could also be investigated. Considering the possible combination of endoperoxides, we would expect additive effects for fasciocidal activity and toxicity, since the pharmacophores, which are also the toxophores, are similar. Regarding the prevention of resistance, a combination could possibly be beneficial if the mechanism of resistance did not involve the active center of the drugs. Future in vitro and in vivo studies should be conducted to investigate this question.

## 4. Materials and Methods

### 4.1. Materials

MT04 and OZ78 were kindly supplied by the College of Pharmacy, University of Nebraska Medical Centre (Omaha, NE, USA, [40]). AS was obtained from Mepha AG (Aesch, Switzerland) and DHA was kindly provided by Dafra Pharma (Turnhout, Belgium). Chemical structures are shown in Figure 1. Triton-X-100 (Sigma–Aldrich, Switzerland) was used as a positive control to induce full cell lysis. Cell culture plates were purchased from BD Biosciences (Franklin Lakes, NJ, USA). Human liver microsomes were purchased from BD Gentest (Woburn, MA, USA). All other chemicals used were ordered from Sigma or Fluka.

### 4.2. Metabolism in the Presence of Human Liver Microsomes

The metabolism of AS, MT04, and OZ78 was tested in the presence of human liver microsomes or heat-inactivated microsomes (70 °C for 20 min, referred to as control microsomes) at 37 °C. AS, MT04 or OZ78 (10 µM each) were added to 1.0 mL of reaction mixture consisting of microsomes (0.5 mg/kg), an NADPH regenerating system, and PBS buffer, pH 7.4. Adding the drugs started the reaction. Aliquots of 100 µL were transferred into Eppendorf tubes after 0, 15, 30, 60, 120 min. The reaction was stopped by adding 300 µL methanol. The samples were then vortexed and centrifuged at 3000 *g* for 30 min. Supernatants were taken for LC-MS analysis.

### 4.3. LC-MS/MS Analysis

LC-MS/MS measurements for AS were performed as described recently [41]. Briefly, sample measurements were done with a high-performance liquid chromatography (HPLC) system (Shimadzu, Kyoto, Japan) connected to an API 365 triple-quadrupole mass spectrometer (PE Biosystems, Foster City, CA, USA) with a turbo ion spray interface. AS was detected in positive ionization mode by selected reaction monitoring (SRM) with a transition of 267.4→163.0. For analyte separation, a reversed phase column was used (2.1 × 10 mm guard cartridge (3 μm) connected to a 2.1 × 20 mm Atlantis T3 3 μm analytical column (Waters, Milford, MA, USA)). A gradient was used (mobile phase A: 5 mM ammonium formiate plus 0.15% (*v/v*) formic acid in ultra-pure water, mobile phase B: 0.15% (*v/v*) formic acid in acetonitrile) at a flow rate of 0.3 mL/min.

The analysis of MT04 and OZ78 [42] was conducted using the same system as described for AS. A phenomenex C8 (50mm × 2.00mm, 5µm, Brechbühler AG, Schlieren, Switzerland) was used for chromatographic separation with a gradient (mobile A: 5 mM ammonium formiate in water and mobile B: acetonitrile) at a flow rate of 0.3 mL/min. MS was conducted in negative ionization mode. For OZ78 and MT04 a mass of 321.4 and 337.4, respectively, was detected.

### 4.4. In Vitro/Ex Vivo Worm Studies

The experiments were performed, as described previously [43]. Adult *F. hepatica* were collected from bovine liver bile ducts obtained from the local slaughterhouse (Basel, Switzerland). After careful washing, a single adult worm was placed per well (6-well plate, Costar, MA, USA) containing 5.4 mL RPMI 1640 (Gibco, N.Y., USA), supplemented with 1% (*v/v*) antibiotics (50 µg/mL streptomycin and 50 U/mL penicillin), 1% (*v/v*) drug solution, and 8% (*v/v*) hemin or FeCl_2_ solution. Adult *F. hepatica* were incubated in the presence of 50 μg/mL AS, MT04 or OZ78 combined with either FeCl_2_ solution or hemin. Control worms were incubated in 1% (*v/v*) DMSO combined with iron or hemin. Experiments were conducted in triplicates and repeated once. Cultures were kept at 37 °C in an atmosphere of 5% CO_2_. The viabilities of treated and untreated worms were monitored at 0, 24, 48, and 72 h. Worms were examined by eye and/or using a dissecting microscope (20× lenses). The viability of adult flukes was scored using the following scale: 4: normal activity; movements were visible without a microscope, 3: reduced activities, 2: very weak activities detected only using microscopic magnification, and 1: death of worm (no movement observed for two min using a microscope).

### 4.5. Cell Culture Conditions

The human hepatocarcinoma cell line HepG2 was purchased from ATCC (Manassas, Virginia, USA). HepG2 cells were maintained in DMEM containing 1 g/L glucose supplemented with 10% fetal bovine serum, 5% non-essential amino acids, and HEPES buffer, pH 7.2 (Invitrogen, Reinach, Switzerland). Cells were incubated under humidified air containing 5% CO_2_ at 37 °C. For cell counting, cells were trypsinized for 5 min until they were detached, stained with trypan blue and then counted using a hemocytometer. Cells were seeded at a density of 20,000 cells/96 well.

### 4.6. Treatment of HepG2 Cells

Stock solutions of the drugs tested were prepared in DMSO 100% (*v/v*). Final concentrations of DMSO did not exceed 0.1% (*v/v*) for all experiments performed. Drug solutions of OZ78 and MT04 in medium tend to precipitate and were, therefore, prepared two days in advance and incubated at room temperature in the dark to ensure complete dissolution. LC-MS measurements confirmed the stability of the compounds (data not shown). Treatments with AS, MT04, or OZ78 (100 or 500 µM) were performed either alone or in combination with 8% (*v/v*) hemin or FeCl_2_ solution for 24 h. The final iron concentration used was 122 µM (stock 1.5 mM) based on previous in vitro studies with *F. hepatica* [36].

The hemin solution was prepared as follows: 5 mg hemin was dissolved in 1 mL of 0.1 M aqueous NaOH, 3.95 ml of PBS (pH = 7.4) and 0.05 ml of 1 M HCl was added to adjust the pH to 7.1–7.4 [36]. For the iron-solution, a hydrated salt was used and solved in ultra-pure water. All solutions were sterile filtered (0.2 µm) after preparation.

In viability assays, HepG2 cells were treated with 0.1% Triton-X or 0.1% DMSO for positive or negative control, respectively.

### 4.7. Adenylate Kinase Leakage (Membrane Integrity Test)

The ToxiLight^®^ BioAssay Kit LT007-117 was obtained from Lonza (Basel, Switzerland). The assay was used according to the protocol supplied by the manufacturer. Adenylate kinase (AK) release was used to assess cell membrane integrity and was determined 24 h after treatment with AS, MT04 or OZ78 with or without the combination of Fe^2+^ or hemin.

### 4.8. ATP Content

To assess cellular ATP content, HepG2 cells were seeded in white-wall 96-well plates with a clear bottom (Nunc). One day after seeding, cells were treated with the respective drugs. After 24 h of treatment, the medium was removed, and the cells carefully washed with PBS containing Ca^2+^/Mg^2+^ (0.1 g/L each) to remove all hemin. Finally, 50 µl PBS was added to each well and incubated with 50 µL CellTiter-Glo reagent as described in the protocol provided by the manufacturer (Promega). Luminescence was measured using Tecan M200 Infinite Pro.

### 4.9. High-Resolution Respirometry

After incubation with different drugs for 24 h, HepG2 cells were trypsinized and resuspended in DMEM with 10% fetal calf serum (FCS) and then centrifuged for 5 min (180× *g*). Cells were resuspended in MiRo5 respiration buffer (110 mM sucrose, 0.5 mM ethylene glycol-bis(β-aminoethyl ether)-N,N,N′,N′-tetraacetic acid (EGTA), 3.0 mM MgCl_2_, 80 mM KCl, 60 mM K-lactobionate, 10 mM KH_2_PO_4_, 20 mM taurine, 20 mM Hepes, 1.0 g/L BSA, pH 7.1). Respiration rates were measured at 37 °C using a high-resolution oxygraph (Oxygraph-2k, Oroboros Instruments, Innsbruck, Austria). Respiration rates were calculated as the time derivative of oxygen concentration measured in the closed respirometer. The amplified signal was recorded in a computer with an online display of the calibrated oxygen concentration and oxygen flux (DatLab software for data acquisition and analysis; Oroboros Instruments). The mitochondrial complex activity was assessed by a standard titration protocol: first cells were permeabilized with digitonin (10 µg/1 million cells) for 5 min. Afterward, for complex I-dependent maximal respiration stimulation, substrates added were glutamate (10 mM) and malate (5 mM), which provide nicotinamide adenine dinucleotide (NADH) to the respiratory chain (complex I activation), followed by addition of ADP (2.5 mM) (state 3, maximal respiration). After a stable signal was reached and marked, rotenone (0.5 µM) was added to inhibit complex I, and then complex II-dependent respiration was stimulated by adding succinate (10 mM), which provides flavin adenine dinucleotide (FADH) to the respiratory chain (complex II activation, state 3). Afterward, complex III was inhibited by antimycin A (2.5 µM), and complex IV-dependent respiration was measured by adding ascorbate (2 mM) and N,N,N′,N′-tetramethyl-p-phenylendiamine (TMPD, 0.5 mM). We confirmed the integrity of the outer mitochondrial membrane by showing the absence of a stimulatory effect of exogenous cytochrome c (10 µM) on respiration [44]. Respiration was expressed as oxygen consumption per 5 × 105 cells.

### 4.10. Measurement of Reactive Oxygen Species (ROS)

Dichlorodihydrofluorescein diacetate (DCF-DA) dissolved in DMSO was used to assess cellular formation of reactive oxygen species (ROS). HepG2 cells were seeded in a black-wall 96 well plate with a clear bottom as described above. Briefly, 30 min before treatment started, DCF-DA was added to the cells at a final concentration of 25 µM. DCF-DA was removed completely before the drugs were added. ROS was measured after 90 min and 24 h of drug exposure using a Tecan M200 Infinite Pro (excitation 485 nm, emission 535 nm).

### 4.11. In Vivo Studies

All animal studies were carried out at the Swiss Tropical and Public Health Institute (Basel, Switzerland) and were approved by Swiss and cantonal authorities (permission: 2070). Female NMRI mice (age: 3–4 weeks, weight: ~20 g) were purchased from Harlan Laboratories (Horst, The Netherlands). Mice were kept in groups of 10 animals in macrolon-cages in environmentally controlled conditions (temperature: ~25 °C; humidity: ~70%; 12 h light/dark cycle) and acclimatized for one week. They had free access to water and rodent diet. For the in vivo studies AS, OZ78 or MT04 were suspended in 7% (*v/v*) Tween-80 and 3% (*v/v*) ethanol.

Since treatment of *F. hepatica* infections in animals is typically performed by a single oral dose [11,45], we performed a pilot study to find out the best time point to kill the mice at which we could observe hepatic toxicity after drug application. This time point was 8 h after drug application (data not shown).

In the final experiment, mice were treated with 200, 400, or 800 mg/kg AS, OZ78, or MT04 by oral gavage. Seven mice were left untreated for control measurements. According to the results of the pilot experiments, where we measured the highest increase in total serum bilirubin, AST or alkaline phosphatase 8 h post-treatment, mice were euthanized by CO_2_ 8 h post-treatment, blood was collected and analyzed as described above. After blood collection, the liver was removed, carefully examined macroscopically, and gross lesions were recorded. Livers were then stored in formalin 10% (*v/v*) until fixation in 4% paraformaldehyde was performed. After embedding in paraffin, sections of 5 μm thickness were obtained, stained with hematoxylin and eosin, and examined using light microscopy.

### 4.12. Statistical Analysis

Results were expressed either as mean ± standard deviation or as mean ± standard error of the mean, as indicated in the figure legends. Means were compared using analysis of variance followed by the Newman–Keuls test to localize differences.

## 5. Conclusions

MT04 and OZ78 have trematocidal activity, which is more pronounced in the presence of iron. Both OZ78 and MT04 are cytotoxic on HepG2 cells, a finding explained best by mitochondrial toxicity due to ROS accumulation. Toxicity and ROS accumulation were enhanced by iron. In mice, single doses of MT04 and OZ78 up to 800 mg/kg were without relevant toxicity, suggesting that these compounds may have a large therapeutic window. Further studies, in particular, long-term administration in healthy and infected animals, are important for estimating the toxicological potential of these new trematocidal compounds.

## Figures and Tables

**Figure 1 ijms-20-04880-f001:**
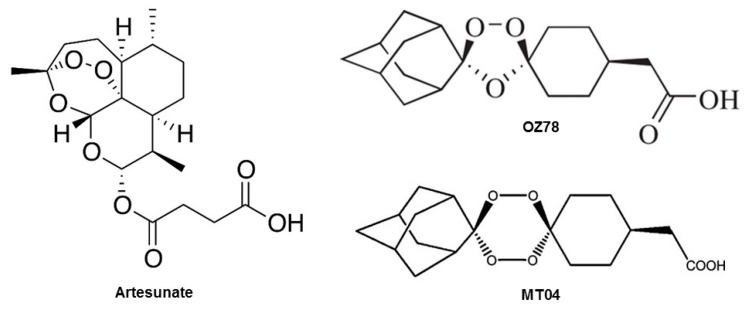
Molecular structure of artesunate, OZ78, and MT04.

**Figure 2 ijms-20-04880-f002:**
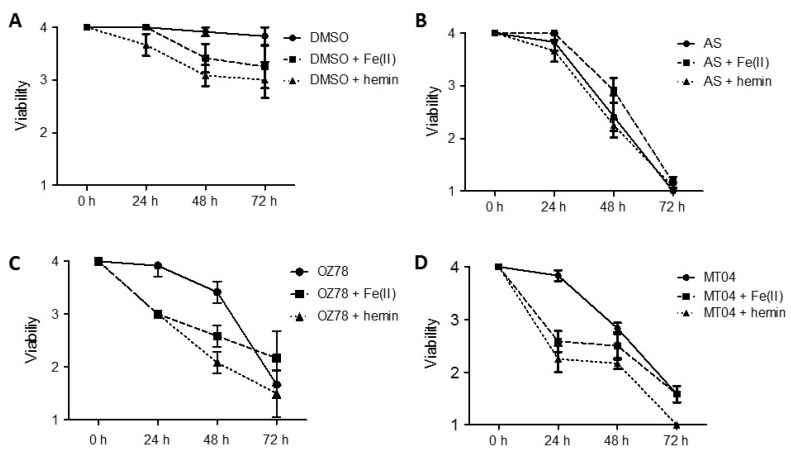
In vitro activity of artesunate (AS), OZ78, and MT04. Adult *F. hepatica* were obtained from bovine livers. Fasciocidal activity was investigated for AS (**B**), OZ78 (**C**), and MT04 (**D**) at a concentration of 50 µg/mL alone (black line) or in combination with Fe^2+^ (dashed line) or hemin (dotted line) and compared to control worms treated with DMSO 1% (**A**). The limits of the whiskers correspond to the standard error of the mean values per time point, *n* = 6.

**Figure 3 ijms-20-04880-f003:**
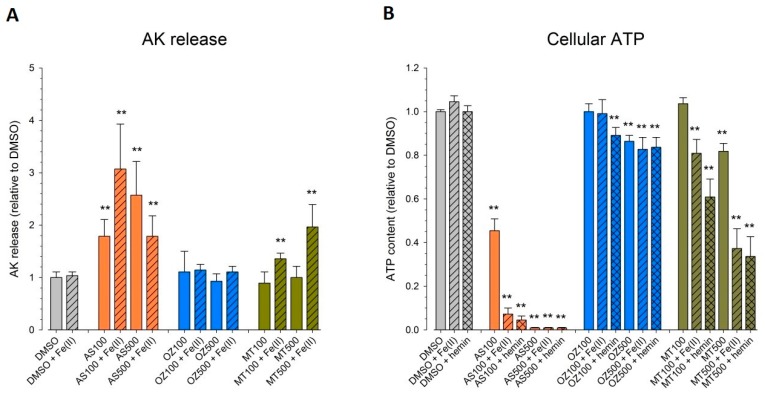
Iron-mediated increase in adenylate kinase (AK) release and depletion of ATP in HepG2 cells treated with AS, OZ78, or MT04. HepG2 cells were treated with AS, OZ78, or MT04 (100 or 500 µM) alone or in the presence of Fe^2+^ or hemin for 24 h. (**A**) AK release expressed relative to DMSO. Triton-X was used as positive control reflecting complete cell lysis (7-fold increase compared to DMSO, data not shown). (**B**) Intracellular ATP content expressed relative to DMSO control. Results represent the mean of five independent experiments carried out in triplicates. Data represent the mean ± STD. Asterisks indicate significance compared to DMSO control, ** *p* < 0.01 vs. control incubations (DMSO).

**Figure 4 ijms-20-04880-f004:**
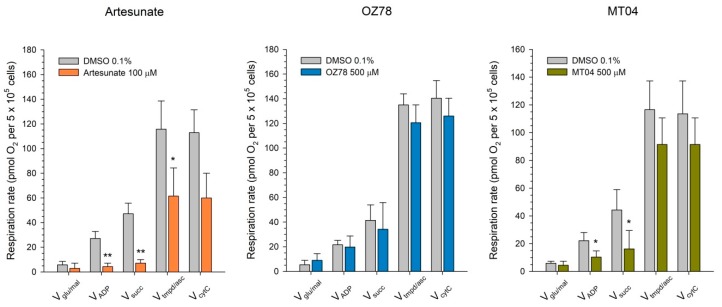
AS and MT04 inhibit different complexes of the mitochondrial respiratory chain. HepG2 cells were treated with AS (100 µM), OZ78 (500 µM) or MT04 (500 µM). After 24 hours, cells were permeabilized with digitonin, and oxygen consumption was measured using the K2 oxygraph (Oroboros). ADP-stimulated respiration rates were compared in the three groups using glutamate (+ malate), succinate (+ rotenone) or N,N,N′,N′-tetramethyl-p-phenylendiamine (TMPD) and ascorbate (+ antimycin) as substrates. Results represent the mean of three independent experiments carried out in triplicates. Data represent the mean ± SEM. Asterisks indicate significance compared to DMSO control, * *p* < 0.05, ** *p* < 0.01.

**Figure 5 ijms-20-04880-f005:**
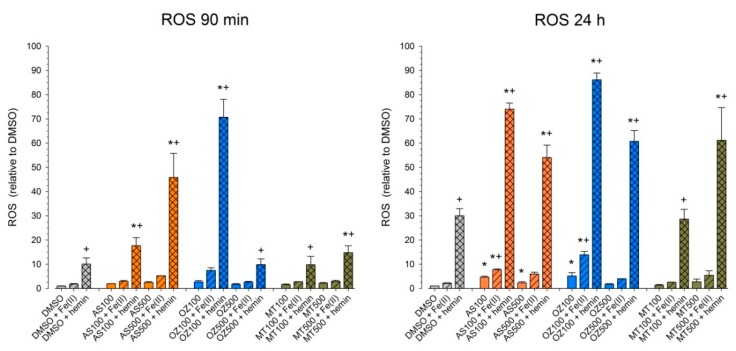
Effect of Fe^2+^ and hemin on cellular generation of reactive oxygen species (ROS). HepG2 cells were treated with AS, OZ78 or MT04 (100 or 500 µM) alone or in the presence of Fe^2+^ or hemin. ROS abundance was measured using dichlorodihydrofluorescein diacetate (DCF-DA) after 90 min and 24 h. Results are expressed as mean ± STD of four independent experiments carried out in triplicates. **p* < 0.05 vs. control incubation containing DMSO, DMSO + Fe^2+^ or DMSO + hemin. ^+^*p* < 0.05 versus the respective incubation containing only the investigated compound but no addition (Fe^2+^ or hemin).

**Figure 6 ijms-20-04880-f006:**
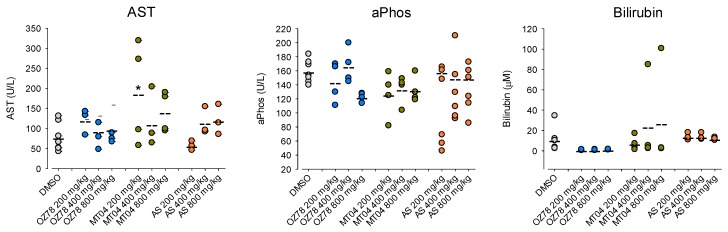
Effect of AS, OZ78, and MT04 on the activity of aspartate aminotransferase (AST) and alkaline phosphatase (aPhos) and on the serum bilirubin concentration. AST, aPhos, and total serum bilirubin were analyzed in plasma from mice treated with single doses of 200, 400, and 800 mg/kg of AS, OZ78 or MT04 or vehicle control (DMSO). Blood was obtained 8 hours after oral administration. Each data point in the graph corresponds to one mouse. The mean of each group is indicated. **p* < 0.05 vs. respective control group (DMSO).

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
