# Peer review of "Exogenous Iron Increases Fasciocidal Activity and Hepatocellular Toxicity of the Synthetic Endoperoxides OZ78 and MT04"

_ijms, 2019, doi:10.3390/ijms20194880_

Round 1

Reviewer 1 Report

The study demonstrates the toxicity and fasciocidal activity of the synthetic peroxides OZ78 and MT04, which have been previously identified as novel anthelmintics. The experimental testing used all appropriate controls and statistical testing throughout. 

1. Given that the concern with current anthelmintics is resistance, perhaps administering both of these drugs simultaneously could help to prevent future resistance. Were any experiments performed to test viability in a combination of the two drugs? It would be interesting to see if they complement each other’s activity. Obviously repeating all of the testing with a drug combination is unreasonable, but some discussion about the potential may be interesting to add.

2. It would be interesting to see if the changes in the chemical structures of the drugs by iron and hemin can be detected by LC-MS, to confirm that the drugs are being altered and the effects are not just complementary. However the authors provide substantial literature support to suggest that this is the case, and the results suggest the same.

Author Response

Given that the concern with current anthelmintics is resistance, perhaps administering both of these drugs simultaneously could help to prevent future resistance. Were any experiments performed to test viability in a combination of the two drugs? It would be interesting to see if they complement each other’s activity. Obviously repeating all of the testing with a drug combination is unreasonable, but some discussion about the potential may be interesting to add.

Answer: This is an interesting question. To the best of our knowledge, studies about the combination of different synthetic endoperoxides or synthetic endoperoxides with artesunate have so far not been published. The reason may be that the mode of action is the same; we would therefore expect an additive effect. Regarding toxicity, assuming that the pharmacophore is also the toxophore, also an additive effect can be anticipated. In our view more interesting is the question whether resistance could be suppressed. This may be so, if resistance is not associated with the pharmacophore, but to other properties of the molecules as for instance transport into the parasites. As suggested, we discuss this point in the discussion section (page 9).

It would be interesting to see if the changes in the chemical structures of the drugs by iron and hemin can be detected by LC-MS, to confirm that the drugs are being altered and the effects are not just complementary. However the authors provide substantial literature support to suggest that this is the case, and the results suggest the same.

Answer: Unfortunately, we did not collect the supernatant of the in vitro experiments described in Figure 2 (killing curves of fasciola hepatica) in order to determine artesunate, OZ78 and MT04 and possible metabolites. On the other hand, it has been described in the literature that Fe2+ and hemin activate artemisinin and fully synthetic endoperoxides to form radical intermediates, which react with hemin, phospholipids and other cell constituents (J Med Chem 2011;54:6443-6455). Since we observed iron-dependent toxicity against fasciola in our investigations, we can assume that this happened also under the conditions used in our experiments. We therefore believe that there is enough evidence in the literature to assume activation and reaction of the compounds used in our studies with cell constituents of fasciola.

Reviewer 2 Report

Fasciola hepatica (F. hepatica) is a foodborne trematode that settles in the liver, bile ducts and gallbladder of the host causing fascioliasis. F. Hepatica affects a large range of people. It has been estimated that 2-17 million people worldwide. Triclabendazole is the first-choice antihelmintic therapy against fascioliasis. However, drug resistance is a threat to treatment and control of fascioliasis in both human and veterinary medicine. The synthetic peroxides OZ78 and MT04 have emerged as potential fasciocidal drugs. However, the effect of iron on fasciocidal activity and hepatocellular toxicity is unknown. The aim of this study was to elucidate the effect of free Fe 2+ and heme complexed iron on the in vitro fasciocidal activity of the synthetic endoperoxides OZ78 and MT04. The researchers also attempted to study the toxicity of these compounds on Hep G2 and on liver morphology and function of mice in vivo. For this study, the researchers used the synthetic endoperoxides OZ78 and MT04, human hepatocarcinoma cell line (Hep G2), adult F. hepatica and female NMRI mice. A quite large number of investigations were used including LC-MS/MS analysis, in vitro/ex vivo worm studies, membrane integrity test, ATP content test, high-resolution respirometry, measurement of reactive oxygen species, in vivo studies and statistical analysis. 

This article is sufficiently novel and interesting to warrant publication. No previous studies were found which investigated the in vitro fasciocidal activity and hepatocellular toxicity of OZ78 and MT04 in the absence and presence of Fe (II) chloride and hemin. 

The authors confirmed that both OZ78 and MT04 are active against F. hepatica and the addition of iron enhances both trematocidal activity and toxicity of OZ78 and MT04. However, no relevant hepatotoxicity was observed in mice. The findings of this study have the potential to improve the management of pseudomyxoma peritonei, which can be expanded in the manuscript. 

The title clearly described the content of the article, while the abstract provided a good summary of the content of the manuscript. In the introduction the authors clearly stated their objectives and the aim of their investigation. The methodology and results obtained were clearly described by the authors. The study design was suitable for the aim of the study with adequate statistical analysis conducted on the results obtained. Appropriate graphs and pictures which were both clear and informative were included in the manuscript. In the discussion, the author summarized the findings with findings being relevant to previous studies. The results obtained supported the claims of the researcher with the speculations and extrapolations being reasonable. There are some grammatical errors in the manuscript. 

Author Response

Fasciola hepatica (F. hepatica) is a foodborne trematode that settles in the liver, bile ducts and gallbladder of the host causing fascioliasis. F. Hepatica affects a large range of people. It has been estimated that 2-17 million people worldwide. Triclabendazole is the first-choice antihelmintic therapy against fascioliasis. However, drug resistance is a threat to treatment and control of fascioliasis in both human and veterinary medicine. The synthetic peroxides OZ78 and MT04 have emerged as potential fasciocidal drugs. However, the effect of iron on fasciocidal activity and hepatocellular toxicity is unknown. The aim of this study was to elucidate the effect of free Fe 2+ and heme complexed iron on the in vitro fasciocidal activity of the synthetic endoperoxides OZ78 and MT04. The researchers also attempted to study the toxicity of these compounds on Hep G2 and on liver morphology and function of mice in vivo. For this study, the researchers used the synthetic endoperoxides OZ78 and MT04, human hepatocarcinoma cell line (Hep G2), adult F. hepatica and female NMRI mice. A quite large number of investigations were used including LC-MS/MS analysis, in vitro/ex vivo worm studies, membrane integrity test, ATP content test, high-resolution respirometry, measurement of reactive oxygen species, in vivo studies and statistical analysis.

This article is sufficiently novel and interesting to warrant publication. No previous studies were found which investigated the in vitro fasciocidal activity and hepatocellular toxicity of OZ78 and MT04 in the absence and presence of Fe (II) chloride and hemin.

The authors confirmed that both OZ78 and MT04 are active against F. hepatica and the addition of iron enhances both trematocidal activity and toxicity of OZ78 and MT04. However, no relevant hepatotoxicity was observed in mice. The findings of this study have the potential to improve the management of pseudomyxoma peritonei, which can be expanded in the manuscript.

The title clearly described the content of the article, while the abstract provided a good summary of the content of the manuscript. In the introduction the authors clearly stated their objectives and the aim of their investigation. The methodology and results obtained were clearly described by the authors. The study design was suitable for the aim of the study with adequate statistical analysis conducted on the results obtained. Appropriate graphs and pictures which were both clear and informative were included in the manuscript. In the discussion, the author summarized the findings with findings being relevant to previous studies. The results obtained supported the claims of the researcher with the speculations and extrapolations being reasonable. There are some grammatical errors in the manuscript.

Answer: Thank you for your comments! We have carefully reviewed the paper and corrected all grammatical errors. We hope that we can meet now the requirements for publishing the article.